# The Impact of COVID-19 on the Performance of Primary Health Care Service Providers in a Capitation Payment System: A Case Study from Poland

**DOI:** 10.3390/ijerph18041407

**Published:** 2021-02-03

**Authors:** Piotr Korneta, Magdalena Kludacz-Alessandri, Renata Walczak

**Affiliations:** 1Faculty of Management, Warsaw University of Technology, 02-524 Warszawa, Poland; piotr.korneta@pw.edu.pl; 2College of Economics and Social Sciences, Warsaw University of Technology, 09-400 Plock, Poland; 3Faculty of Civil Engineering, Mechanics and Petrochemistry, Warsaw University of Technology, 09-400 Plock, Poland; renata.walczak@pw.edu.pl

**Keywords:** COVID-19, primary health care, performance, capitation payment

## Abstract

In Poland, as in many other countries, the use of capitation payment schemes in primary health care is popular. Despite this popularity, the subject literature discusses its role in decreasing the quality of primary medical services. This problem is particularly important during COVID-19, when medical entities provide telehealth services to patients. The objective of the study is to examine the effects of COVID-19 pandemic on the performance of the primary health care providers in Poland under a capitation payment scheme. In this study the authors use data from interviews with personnel of medical entities and financial and administrative reports of primary health care providers in order to identify how this crisis situation impacts the performance of primary health care entities, under capitation payment system. The performance indicators include both the financial and quality measures. Selected to the case study primary health care service providers significantly improved their profitability due to considerable costs savings and reduction of services provided to patients in a time of COVID-19 pandemic. Capitation payment system proved to be inefficient, in the studied pandemic period, in terms of the services provided by primary health care service providers to patients and the funds paid to them, in exchange, by the government entities.

## 1. Introduction

Primary health care is often described as the foundation of a strong health care system [1]. According to the World Health Organization (WHO) “All people, everywhere, deserve the right care, right in their community. This is the fundamental premise of primary health care” [2]. Originally fundamentals of the Primary Health Care (PHC) were presented in the Declaration of Alma-Ata developed during the International Conference on PHC in 1978 organized by the WHO and the United Nations Children’s Fund (UNICEF). Based on this declaration, all countries should develop a PHC policy and a comprehensive national health system in order to keep people healthy. Public health care will help solve most health problems in the society. It might be achieved by “better use of world’s resources” [3]. Publicly financed health care will help improve the health of the population, give “health to all”, and fight widespread diseases, for example a pandemic. According to the Pareto law, it is possible to solve 80% of health problems using only 20% of funds. The main goal is to wisely utilize these 20% in order to achieve the best, optimal effect. Since 1978 Alma-Ata declaration many initiatives regarding PHC have emerged. WHO, year by year, prepares guiding principles regarding national PHC strategies [4,5,6,7], however the WHO claims that there are many countries that spend too little for society health and that money is spent inefficiently. The WHO has declared new, improved PHC aims associated with sustainable development goals [6]. Quality and economy are declared as very important factors of the new strategy [8].

Improving the quality of primary care should be the primary goal of any government’s health policy. The goal of primary health care in Poland is to provide all persons entitled to benefits with comprehensive and coordinated health care services at the place of residence. These services are provided in public health centers, primary clinics, doctors’ offices, and in medically justified cases, also in patients’ houses and at social welfare homes [9]. Polish primary health care system consists of around 12 thousand primary health care service providers (PHP), who provide services to around 3 thousand patients each. The institution established to finance health care in Poland is the National Health Fund (NHF).

PHC and primary care physicians, commonly referred to as general practitioners (GP) or family physicians, must be health insurance physicians, those who have a contract with the NHF for the provision of health care services in a given calendar year or work in a facility with which the NHF signed an agreement. Either public or private facilities might receive NHF contract. The functioning of primary health care is based on the right to personal selection of preferred PHP and their GP. The patients choose a primary care physician themselves, regardless of the place where the services are provided and the place of residence. The choice is made by submitting an appropriate declaration, indicating the choice of the doctor by whom the patient would like to be treated, as well as the primary care nurse and midwife. Availability to a specific PHP is limited by the number of patients per doctor under the contract, according to the guidelines of the NHF, it should not exceed 2750 patients. Following the signature of patient’s access form to PHP, the selected primary care physician receives an annual capitation payment per patient who is registered with a doctor. The patient is allowed to choose a primary health care doctor, primary health care nurse and primary health care midwife free of charge two times during the calendar year. Until December 2020, more frequent changes were charged €18.

The services of GP and primary health care nurse include comprehensive medical care for a person, family, community in the living environment, taking into account the place of providing services. Available services of the GPs and nurses in Poland are presented in Table 1.

Then national health care system is financed from obligatory 9% health care payment levied on all employment contracts in Poland. Collected money is next transferred to NHF, who is responsible for its administration and further distribution, including the PHC.

Primary health care in Poland is financed based on the capitation payment system that is based on the multiplication of the number of participants registered in the health facility by the capitation rate per person [10]. Under the capitation payment, providers receive an agreed sum of money for each patient registered for a specified period of time to provide him with predetermined services, with the expectation that the capitation payment will favor the efficient use of limited health care resources by controlling the volumes of services delivered and associated costs. Under a capitation payment scheme, both payers and health care providers know their budgets in advance [11].

The capitation model assumes payments calculated to cover the remuneration of a multidisciplinary clinical team, infrastructure costs (e.g., the costs of implementing electronic health records) and other expenses deemed necessary for the operation of a primary care facility [12]. The NHF payment must cover the costs of medical advice, tests and additional services. In case of more expensive tests, such as computed tomography or magnetic resonance imaging, they are separately contracted by the NHF and are not included in the cost of advice.

Capitation payments are tied to the number of patients assigned to a clinic, and take into account factors that are believed to increase the demand for primary care services (age, chronic conditions). Capitation payment vary according to the age group and changes every few months. All values were converted from Polish zlotys into EUR using the average European Central Bank exchange rate of 4.4524 for the period between 1 March to 30 November 2020. Monthly capitation payment (presented in the Appendix A, Table A1.), for a child up to 6 years of age, PHP receive €8.64 per month, and for each adult aged 40–65-€4.29 per month. With patients age the subsidy increases. For patients aged 66–75 PHP are granted €8.71, and €9.92 for the oldest patients. The payment for people with chronic conditions is even higher and amounts to € 10.24 per month. Nursing care is also financed, for the youngest children and the oldest patients PHP receive €1.56 for each patient, the payment for the remaining patients is €0.78 per month per patient. For patients registered with the midwife PHP receive €0.52 per month per declared patient [13,14]. Other components of general practice funding in Poland include school nursery services under several NHF programs. These programs are also granted on a per capita basis.

The main reasons for introducing the capitation payment system is usually a cost reduction, disease prevention promotion, ensuring access to health care and improving the quality of patient care and health outcomes [15]. The remuneration of GPs in Poland is, based only on capitation scheme and does not contain other elements, e.g., fee-for-service (FFS), that could motivate them to pro-quality activities, e.g., to improve the efficiency of their work. The capitation payment scheme is popular in many European countries as a cost control tool, but the problem is that only in Poland this is the only payment method in primary health care. In other countries, the payment system is a combination of both capitation and FFS (Belgium, Denmark, Ireland, Luxembourg, Romania and Slovenia), capitation and payment for results (P4R and/or global budget (Estonia, France, Hungary, Latvia, Lithuania, Portugal, the Netherlands, Spain and the United Kingdom) [16]. The current trend is to introduce multiple payment methods for primary health care to achieve multiple goals for access, quality and efficiency [17]. Many countries in Europe have already taken the steps to adjust and combine the capitation and other payment mechanisms that encourage the delivery of high-quality health care. According to the authors, the reforms undertaken in the EU’s primary health care systems are warranted and such steps should also be taken in Poland.

Capitation payment system encourages GPs to register more patients to their primary health care clinics, because the more patients they have on their lists, the higher their financial reward is. However, there is no mechanism encouraging referring patients to specialist treatment or diagnostic tests. Indeed, there may appear perverse incentives for the managers of primary health care clinics to decrease the quality performance based on access indicators because the more the services provided, the less profits available for the clinic. 

The new coronavirus (COVID-19) pandemic, caused by the SARS-CoV-2 virus, has required enormous changes to everyday life to reduce transmission, morbidity and mortality [18]. This disease resulted also in a significant change in the functioning of primary health care. Along with the change of health care procedures, it is necessary to change the assessment of their quality and availability. Work in the post-COVID-19 reality cannot be planned on the basis of previous solutions. New data, new health care use must be taken into consideration instead of running the business as usual [19]. First of all, new rules and regulations and new standards for maintaining the quality and availability of medical services are necessary, which will allow PHP to adapt faster to the changing environment.

Especially now, with the COVID-19 pandemic the question arises how do the capitation payment models of primary care affect the costs, financial performance, supply and quality of medical services, the number of patients served and diagnostic tests performed, which are determinants of continuity and quality of patient care?

The evidence from the research conducted so far suggests some positive effects, like cost containment and income improvement following the introduction of capitation payment scheme [20,21,22,23,24,25,26,27,28,29,30,31]. It has been also indicated that the capitation system does not increase the overall costs [32]. There are some negative indications regarding the capitation system provided in the literature though. These indications relate primarily to the quality of performance and access to medical care in health care entities. There are also studies postulating that medical entities functioning under capitation payment scheme tend to decrease the quality and quantity of their services [28,33,34,35,36,37]. Finally, several researchers claim there is no negative impact of capitation payment system on productivity and quality performance [12,33]. To our best knowledge, no results of a study, on the impact of crisis situation, such as COVID 19, on the quality of performance in primary health care entities under capitation payment scheme, have been provided so far. 

The objective of the study is to examine the impact of COVID-19 pandemic on the financial and quality performance of the primary health care providers in Poland under a capitation payment scheme. It was checked the influence of COVID-19 on an access to care, levels of treatment activity, the finances of a primary care clinics and the quality of care assessed by managers of primary care working in the capitation payment system. This study is the first to explore the relationship between practice-level funding based on capitation scheme and behavior of GPs practices during the COVID-19 pandemic. This study uses financial and administrative data at the health care provider level. The study used administrative and financial data obtained from the accounting departments of the for four Polish clinics collected in the period from March to November 2019 (pre-COVID-19 period) and from March to November 2020 (during COVID-19 period). In order to better understand these data the interviews with managers and medical staff of the clinics have been organized.

Capitation financing of medical services in PHC was introduced in Poland in the mid-nineties of the last century. After many years of functioning, it is fairly common to believe that this financing system is not used as a tool to encourage GPs to work more efficiently and to provide high-quality patient care. It is also not considered a fair way of remunerating physicians, as it does not reward them adequately to the cost and work involved in providing medical care. Besides, it does not treat patients in a way that corresponds to their actual health needs. [38]. The fundamental question is how this financing system works in a crisis situation, such as COVID-19, when the health care system should be particularly effective and ensure that resources are distributed fairly according to patients’ health needs. The results of the study should therefore be interested in the stakeholders of the central level of the health care system, especially the Ministry of Health and NHF.

## 2. Materials and Methods 

### 2.1. Study Design

The research design was a descriptive study. A case study was used to describe and compare the performance of the primary health care providers in Poland operating under a capitation payment scheme before and during COVID-19 pandemic. Data was collected using financial statements and other sources of administrative data of the selected four primary health care entities. This method incorporated also financial and medical data analysis, interviews and direct observation.

Four Polish typical primary health care service providers were selected for the present case study, those were clinics which met the following criteria: (1) have been providing primary health care services for at least 5 years; (2) had between 3 to 6 thousand of affiliated patients to GP (adults and children); (3) whose majority of sales revenues came from NHF. The selection was limited to the clinics that operate only in primary health care sector, as the vast majority of Polish primary health care clinics provide auxiliary services, with almost none of them operating solely in primary health care sector. They are also predominantly financed by the NHF. A description of four primary health care service providers selected to the case study are presented in Table 2.

All clinics provided 45,426 services to patients during 9 months in 2019 and 42,779 services during 9 months in 2020 (a mean monthly value of 5046 and 4756 respectively). Polish clinics are more and more willing to implement coordinated health care. They take care to comprehensively approach the patient and ensure continuity of treatment, as well as have internal supervision over the quality of services. Pursuant to the Act of 27 October 2017 on primary health care, GP is the patient care coordinator and is responsible for cooperating with a nurse and midwife, as well as with other specialists who look after the patient [39]. The goal of coordinated care in Poland is: better patient-centered care, increasing the role of prevention and health education; optimization of the treatment process by improving the organization and functioning of primary care; increasing the efficiency of primary health care and coordinating activities at different levels of health care as well as ensuring continuity of treatment and internal supervision over the quality of services in primary health care [40]. However, as outpatient specialist care funding is based on a fee for service and, to a limited extent, on the basis of homogeneous patient groups (DRG), data in this regard was not included in the analysis.

### 2.2. General Practice Financial and Administrative Data 

The study used administrative and financial data collected in the period from March to November 2019 (pre-COVID-19 period) and from March to November 2020 (during COVID-19 period). 

Since the amount of the capitation payment is based on the number of insured patients affiliated to the provider, the revenues of primary health care service providers in Poland rely largely on the per capita ratio and the number of registered insured patients. For analyzed period, the authors calculated medical margin on the base of capitation payment for each practice as amount of money per registered patient and medical costs. 

The dataset considered in this article contains data at PHP level regarding NHF payments, costs of remuneration and medical margin. Those costs were compared to administrative data. According to the authors, access to data at the level of each clinic is the strength of this study. In other studies, the lack of data at the single clinic level is considered a limitation. It often results from the fact that multi-clinical health care entities prepare consolidated cost reports that do not contain financial data for each clinic separately. These systems report administrative data separately for each clinic and a single set of medical margin financial data for all clinics. In such a situation, it would then not be possible to compare the changes in terms of qualitative and financial achievements for each clinic separately [41].

Financial and administrative data for four clinics were obtained from the accounting departments of the clinics. Data for the period of March to November 2019 (pre-COVID-19 period) was compared to the data of March to November 2020 (during COVID-19 period). The financial variables used in the analysis are medical margins calculated on the basis of sales, salaries of medical staff, diagnostic and medical materials costs. Variables that the authors considered to be associated with quality health care performance are based on the productivity measures and regard patient admissions by GPs (site services, home visits, telemedicine), number of selected services provided by nurses (ECG, Holter, injections, spirometry, vaccinations, flue vaccinations) and laboratory diagnostics (biochemistry, hematology, hormones, urine and feces tests, conclusion system, cancer diagnostic, autoimmunology).

### 2.3. Data Analysis

A horizontal analysis evaluating the relative changes in different variables over time were performed. The study was executed to analyze changes in financial and quality performance of the clinics after COVID-19 pandemic outbreak. A case study analysis was used to compare the changes in performance indicators in two periods (pre-COVID-19 and during COVID-19). This analysis takes into account changes in financial measures-revenues, costs and medical margins (financial performance) and productivity measures-based on health services performed (quality performance). Initially, the difference between the data from the first and the second period was calculated, then the percentage change over time was quantified. The analysis was carried out at each primary health care service provider level. Analysis was performed using Microsoft Excel 2016 (Microsoft, Redmond, WA, USA) to examine changes in value and structure regarding health care utilization and financial indicators.

The interviews with managers and medical staff were used to corroborate the patterns that evolved from the financial and administrative data collected, so that the validity of the findings, changes of the situation of PHP and behaviors of medical staff could be enhanced and better understood.

## 3. Results

### 3.1. Performance Analysis of Clinic 1

Mean monthly numbers of patients affiliated to the Clinic1 in studied periods are presented in the Appendix A, Table A2. The sales revenue breakdown, expenditures and medical margins calculation for the first of the studied clinics are provided in Table 3.

As presented in Table 3 sales revenue and medical margins of the Clinic 1 have grown in the period of pandemic by 76.5% and 153.9% respectively. Such phenomenal outcome had three major drivers. Firstly, NHF increased the amount of fees paid per patient. Secondly, the clinic introduced paid out of pocket SARS-CoV-2 test (presented as diagnostics in Table 3). The launch of this new product was a key sales and profitability driver. Thirdly, the operational activity of the clinic significantly decreased during the time of pandemic.

The manager of the clinic said: GP provided less services to patients, so she reduced the number of overtime hours as compared to the prior period. This is visible in Table 3, as the salaries of doctors serving adult patients decreased by 17.6%. Next, she said the schools have been closed since April 2020, so all school nurses provided no services. This fact has not influenced monthly fees paid by NHF from public money to the Clinic. She transferred school nurses to SARS-CoV-2 testing facilities. As a result the school nurses were financed by NHF and provided commercial services to patients. Finally, she added: the number of home visits provided by midwife, who is paid on hourly basis, have decreased significantly.

Table 4 presents the number of services provided by medical doctors and nurses to patients of the Clinic 1, including home visits and telemedicine services, provided by the phone in the periods March–November 2019 and March–November 2020.

As presented in Table 4, the total number of services provided to patients, despite the launch of telemedicine visits, decreased during the pandemic by 9.6%. The clinic nearly ceased provision of home visits to immobilized patients. Finally, one shall note the role of telemedicine was increasingly growing with 751 services provided to patients only in November 2020 (36% of all medical doctors services provided in November 2020). The number of services selected to the study provided by the nurses of the Clinic 1 during the pandemic has decreased sharply by 43%.

### 3.2. Performance Analysis of Clinic 2

Table 5 presents sales revenue breakdown, expenditures and medical margins calculation for the second of the studied clinics. Monthly mean numbers of patients affiliated to the Clinic 2 in studied periods are presented in the Appendix A, Table A2.

Despite the decrease of the number of patients affiliated to the clinic, its sales revenue and medical margins have grown by 9.3% and 45.5% respectively. According to the manager of the clinic, this outcome results from the increased fees paid by NHF per affiliated patients to the clinic, and considerably lower operating activity of the clinic. He has contracted significantly lower number of medical doctors, nurses and midwife working hours, with the latter providing almost no services to patients in the time of pandemic.

Table 6 presents the number of services provided by GPs and laboratory diagnostics provided to patients of the Clinic 2 in the periods March–November 2019 and March–November 2020. Due to IT system insufficiencies, the Clinic 2 could not deliver requested information regarding the number of services provided by the nurses for patients.

The number of services provided in the clinic decreased slightly by 1.5%, while the number of home visits reduced significantly by 63.2%. The manager regrets that he has not introduced telemedicine services until September, as these services compared to those held at the premises of the clinic, last shorter, so more patients receives and advise within an hour and, additionally, these kinds of visits are welcomed by medical doctors. In November, almost one third of medical doctor services were provided by the phone. As presented in Table 6 the monthly mean number of laboratory diagnostics has decreased significantly to 6,391 tests by 39.4% during the pandemic. The results presented in Table 6 are aligned with the reduced diagnostic costs presented in Table 5.

### 3.3. Performance Analysis of Clinic 3

Table 7 presents sales revenue breakdown, expenditures and medical margins calculation for the third of the studied clinics. Mean monthly numbers of patients affiliated to the Clinic 3 in studied periods are presented in the Appendix A, Table A2.

As presented in Table 7 sales revenue and medical margins have grown by 21.2%, and 41% during the period under consideration, despite the decrease of the total number of patients. The increase of diagnostic sales results from temporary SARS-CoV-2 tests which the company sold during two months of pandemic. 

According to the manager, due to appearance of a new competitor in a close vicinity, the clinic’s premises and resources turned out to be excessive. The clinic recognized extra margin on midwife services, who is billed on an hourly basis and provided very little services during the pandemic. The manager was asked if the number of medical doctors working hours was also reduced. He claimed that the reduction was insignificant, because the two medical doctors, who provide the bulk of services to adults are employed full-time. Hence, it was not possible to reduce their working hours, they were working at a normal pace.

Table 8 presents the number of services provided by GPs, nurses and the number of laboratory diagnostics provided to the patients of Clinic 3 in the periods March–November 2019 and March–November 2020.

The manager of the clinic stated that the doctors did not want to provide services in patients’ homes, additionally patients were afraid to ask for the doctor’s home visit as not to get infected with COVID-19. This belief of patients comes from the fact that medical doctors have many contacts with various patients so they are carrying higher risk of infection transfer. Home visits last three times longer than visits at the premises of the clinic, so Clinic 3 had contracted full-time medical doctors and could provide more services within contracted time. The manager added that he was surprised how efficient telemedicine services turned out to be, especially as one of medical doctors was currently 69 years old. He never imagined that that elderly doctor would prefer telemedicine visits to site services.

The number of services selected to the study provided by the nurses of the Clinic 3 during the pandemic has decreased by 13.1%, mostly because of lower number of ECG and injections. The manager of the clinic stated that recently the interests of patients in any vaccinations has grown.

As presented in Table 8, the number of laboratory diagnostics have decreased during the pandemic by 15%; however, data related to November 2020 is the most interesting During the last month of studied period, the Clinic 3 provided only 154 diagnostics; in comparison to the mean monthly number of diagnostics before the pandemic of 1030, this indicates that the diagnostic role of PHP has nearly vanished. The nurse who was also interviewed presented the sources of such decline. She confirmed that, according to her perception, the number of diagnostics has been reduced greatly during recent period. The main reason was patients’ fear of infection. The patients simply delayed doctor or nurse appointments. They came to a doctor or a nurse, only if they had to, i.e., when they could not cure themselves on their own. Secondly, she noted, the patients tended to receive more advice by phone, and did not came either to the clinic nor for a diagnosis. Thirdly, she commented the sharp decline of diagnostics in November 2020 was due the absence of one of two nurses due to the COVID quarantine.

### 3.4. Performance Analysis of Clinic 4

Table 9 presents sales revenue breakdown, expenditures and medical margins calculation for the fourth of the studied clinics. Monthly mean numbers of patients affiliated to the Clinic 4 in studied periods are presented in Appendix A, Table A2.

Although the number of patients of the Clinic 4 was reduced by 935 (10%) through the pandemic, its sales revenues and medical margins have decreased only by 2.9% or 2.4% respectively. The manager was asked about that significant reduction in the number of patients. She stated that this was not a problem of the pandemic but the reason was competition that appeared. In the close vicinity of the clinic there were two other PHP. One was smaller than Clinic 4, while another was larger and located nearby; it possessed about 6 thousand affiliated patients. At the beginning of 2020, this larger competitor had contracted a medical doctor from the Clinic 4, offering him considerable salary increase. Together with the medical doctor, the number of patients was reduced. Patients decided to change their PHCs hand in hand with their doctor.

Despite that, the financial results have not deteriorated, mostly because NHF increased per capita payment, and because of lower diagnostics cost.

Table 10 presents number of services provided to the patients by the GPs and nurses in March–November 2019 and March–November 2020.

The number of services provided by the medical doctors of the clinic reduced significantly by 18.4% during the pandemic. Clinic 4 has not introduced telemedicine on a larger scale. The manager of the clinic claimed that doctors were not willing to provide services by phone, and that it was difficult to decide whether a patient should come to the clinic or receive an advice remotely. Patients used to call to the receptionists who were not medical doctors, and did not have proper medical knowledge to assess whether a patient should come or use a telemedicine service. Two medical doctors were asked why they did not use phone services. It turned out that that the diagnostic part of the service was a major problem for them. By phone, they could not see the throat and tonsils, they could not auscultate the heart and lungs, and they evaluated such kind of advice as a risk. They were afraid that they might miss significant symptoms, which could be only noticed during the normal visit.

As presented in Table 10, the number of services provided by the nurses to the clinic patients reduced considerably by 32.8% through the pandemic. The decrease was especially visible in the last of studied month, i.e., November 2020 when the number of infections in Poland was the highest.

Clinic 4 did not provide information regarding laboratory diagnostics since their IT system had limited functionality. The interviewed nurse discussed laboratory diagnostics. She said that also the daily number of blood donations taken in the treatment room had decreased significantly, more than a half.

## 4. Discussion

This paper provides further evidence for the relationship between general medical practice funding and the financial and quality performance of primary health care. This study shows that during a crisis situation such as the COVID-19 pandemic, capitation funding is associated with the improvement of the financial situation but also with a deterioration in the efficiency of the primary health care practice assessed on the basis of admission rates and the number of medical services. The earlier studies have already shown the impact of the crisis on the financial indicators of primary health care entities financed under the capitation scheme [32]. However, it is the first study that show the decrease of access indicators in primary health care entities in a crisis situation such as COVID-19.

The main objective of this study was to assess the impact of the COVID-19 pandemic on the performance of primary health care facilities financed, under the capitation payment scheme. The main purpose of introducing this payment scheme is usually to contain the primary health care costs in a fixed fee, which is next to be controlled by the number of affiliated patients to the primary health care service providers. The study found that primary care health service utilization and related expenditure fell after the COVID-19 pandemic outbreak. Thus, the functioning of the capitation payment scheme led to an improvement in the financial results of the primary health care facilities, but in the same time the quality of performance measured with access quantities decreased.

The effect of payment models on various aspects of primary medical practice has already been a subject of extensive studies in the literature e.g., [24,25,26,27]. Research literature suggests that health care providers are adapting very quickly to current payment models to ensure the profitability of their practices.

A key objective for introducing capitation payment method in many countries is to control utilization and contain the cost of claims paid by the national payer [11]. Other researchers have also already shown that this payment method of health care providers doesn’t affect their potential loss of income during a crisis. GPs and health care providers, who are paid mainly under a capitation scheme, are less susceptible to various types of shocks than those who are largely paid on an activity basis e.g., FFS or P4R. For service providers financed under activity-based schemes, the crisis can seriously disrupt income flows. Therefore, countries with such payment systems should use different strategies to offset the income loss of primary care providers due to reduced demand for health services and additional COVID-19 related expenditure [24].

The study presented in the present paper confirmed that health care providers financed under the capitation scheme have no problems with the loss of income in crisis conditions. Moreover, it shows that a pure capitation system (Poland is the only country in Europe operating in such system) leads to an improvement in financial results, which are not justified by an increase in efficiency. The findings regarding COVID-19 pandemic are also in line with the findings of other researchers who have shown that a capital payment scheme leads to reductions of health care expenditure for services that have to be paid by providers, i.e., for which providers do not receive additional fee than the capitation one. Therefore, these reductions are a key source of profitability improvement for health care service providers [28,29,30].

Payment systems should take into consideration financial outcomes and value and quality of primary health care delivered to patients. Also, the main purpose of the capitation payment scheme should be to contain the costs while ensuring the quality of service provided to insured patients. In some countries capitation payment has been piloted not only as a mean of cost containment, but also to induce competition between providers to improve responsiveness to health care services delivery [33]. However, it was noticed the changes in payment systems often attract responses from suppliers, which may negatively affect the quality and quantity of their services [33]. While the literature has found that the capitation payment encourage efficiency [31], drives down the costs [29,30] and serves as a key source of income for service providers, it has been also noted that it reduces the quantity and quality of medical care provided [28], encourages to withdraw from treating high-risk patients and negatively affects the patient-provider relationship [34].

In theory, the capitation payment system of primary health care serves as a cost control tool, shifting some financial risk to service providers, thereby encouraging them to provide care more cost-effectively. So far, in the literature, it has been emphasized that in the non-pandemic periods financial risk imposed by this payment scheme may encourage service providers to under-supply health care services or to provide sub-standard medical care to reduce costs [36]. Furthermore, due to inadequate controls and weak monitoring mechanisms, capitation payment scheme may lead to other unintended consequences, such as service providers requesting informal payments from insured customers to offset some of the risk [33]. The capitation payment system encourages primary health care service providers to operate effectively and efficiently, but mostly in terms of financial performance indicators. This system doesn’t motivate promotional and preventive activities.

The pure capitation scheme has several weaknesses and is not always in line with today’s health care system priorities. While capitation is a better payment system to control costs, it can lead to a selection of patients requiring fewer services. Therefore, the adopted pure capitalization scheme is not well suited to meet the challenges of an aging population and the increasing burden of chronic diseases [17]. The studies conducted in other countries have already found that the capitation method encourages providers to under-produce health care services [35]. It was also proved that it is not suitable during a crisis situation.

The question therefore arises whether the other payment scheme e.g., FFS (fee-for-services) payment scheme should not be the preferred method of payment in insurance systems. Conceptually, the capitation method is expected to favor underproduction and the FFS method is expected to favor the overproduction of health services in insurance systems.

It has been already noticed in a literature that compared to FFS, introduction of capitation model results in less patient visits, lesser continuity of care, but patients are more satisfied with access to a doctor [42]. Replacing FFS payments with capitation models most often leads to a smaller number of medical services, not only in the area of primary care but also in area of ambulatory care, and to fewer elective surgeries (e.g., cataracts and tubal ligation) [43]. FFS is typically associated with more primary care visits/contacts (about 5-7% more compared to capitation), as well as more specialist visits and more diagnostic and therapeutic services. Moreover, the capitation payment model theoretically motivates GPs to serve healthier patients and reduce the scope of services [44]. It has been also estimated that the transition to capitation payment in primary health care leads to an average of 3% fewer laboratory requisitions per patient in the short term [45]. Traditional forms of FFS and capitation remain the most common payment methods for primary care in the EU health systems. However, they would need to be adjusted (for example through risk-correcting capitalization payments) or combined to best meet growing health care needs.

In the literature, there are opposite opinions regarding the quality of medical services and productivity in primary health care in a capitation payment system. According to Meredith and Rosenthal [12], although primary care physicians are not paid for additional costs such as referrals to specialists and diagnostic tests, the capitation scheme of payment includes significant performance incentives in terms of quality and cost-effectiveness (ranging from 15 to 25% of total payments). Also, a study that examined the impact of capitation payment on the perceived quality of health services and the prevalence of out-of-pocket payments in Ghana showed that this payment scheme was associated with a greater likelihood of out-of-pocket payments but no differences in perceived service quality were observed [33].

The results of this study confirm the evidence from the literature that the capitation method of financing primary health care may lead to the reduction of health care services [37]. Additionally, it shows that this is especially noticeable in crisis conditions because the improvement of financial results in the capitation system takes place in a situation of reduction of medical services.

Considering that diagnostic tests ordered by GPs do not bring direct income in the capitation payment scheme, but only generate additional costs, the question arises how to improve this payment model so that it motivates service providers to commission more tests. According to L’Esperance et al. [46] the quality of primary care depends on the value of capitation funding per patient. Higher capitation funding is associated with higher levels of primary care quality. This includes all measured dimensions of the quality of care (patient safety, patient experience, clinical effectiveness). The other studies suggest that insurance systems could achieve better results through integration of capitation payment systems with effective monitoring and regular health care provider inspection can be helpful in preventing reduced patient admissions and the number of tests performed. An appropriate active monitoring mechanism could encourage providers to provide patients with adequate health care [47]. The other possibility is that the capitation payment per enrollee should be revised to conform to economic circumstances. Also alternatively, more innovative payment methods could be used to encourage coordination of care and improve the delivery of care for patients with chronic conditions, e.g., add-on payments, P4R schemes, bundle payments for particular patient groups [38].

## 5. Conclusions

The study found that that during the COVID-19 pandemic, the operations provided within the capitation payment system are less efficient, while the financial results of primary health care service providers improved in comparison with the non-crisis circumstances. The results indicate that the less patients served and the less medical services performed, the better financial performance of primary health care service providers. It was found that the considerable reduction in the number of procedures performed in all practices of studied clinics in the period Mar–November 2020 following the COVID-19 pandemic. The significant reduction of activity of the studied clinics lead to considerable costs reductions. Given the capitation fee has not been reduced but increased slightly, in line with historical increases, the revenues of primary health care service providers increased. The reductions of costs and the growth of revenues lead to significant profitability improvement in the time of pandemic for primary health care service providers. 

The results obtained in this study showed that the crisis situation such as the COVID-19 pandemic might affect the attitudes and the performance of primary health care service providers in a capitation system of payment. It was identified that the number of patients admitted to medical doctors, nurses and midwifes in the time of pandemic has been reduced. It was observed the same tendency for diagnostic tests, which number, has been also reduced considerably in a COVID-19 time. These declines result from several reasons, such as: (1) patients’ fear of infection, (2) reduced supply of medical staff which have a higher risk due to numerous contacts with patients’ fear of infection, (3) a tendency of primary health care service providers to limit clinical activity as to increase profits. This shift of attitudes may affect the perception of the quality of services by patients. 

The study revealed that the medical margin in studied primary care facilities surveyed increased significantly after the COVID-19 pandemic outbreak. Three out of four primary health care service providers increased medical margins by: 153.9%, 45.5% and 41.0% respectively. The medical margin of the remaining clinic fell by only 2.4%. This was due to the considerable reduction of the number of patients, which mean number in studied period fell by 10%, resulting from a severe competition in its region. Such considerable improvement of medical margins was because of two major reasons. The first, is the capitation payment mechanism itself, which caused that the medical revenues remained unchanged. The second is a considerable reduction of costs, primarily relating to the salaries of medical staff, but also the diagnostics test, resulting from reduced activity of primary health care service providers in a pandemic.

The results of this study indicate that the primary health care service providers, operating under capitation system mechanism, are the only ones who significantly benefited, in financial terms, from COVID-19 pandemic. Meanwhile, the community (i.e., patients) did not receive services as needed with appropriate quality, while the government paid a full charge to primary health care service providers. This finding may be of particular interest to policy makers. During the pandemic, the government revenues tend to decrease significantly due to lower taxes resulting from reduced activity of the companies, while the other branches of health care industry become more expensive, as the health care system must finance the increased costs of diagnostics (COVID-19 tests) and the increased costs of infected patients’ treatment. Linking the payment systems of primary health care with the services provided instead of with the number of patients, would allow, in crisis situations such as COVID-19, to direct the funds where they are needed, instead of overpaying primary health care service providers.

This study has several limitations. The study was conducted in only four primary health care service providers in one country. It is recommended to follow up the results with research on larger scale, comprising of other regions. Next, it should be noted that the quality of services assessment is based mostly on administrative and financial data and does not include patient surveys. This study did not investigate, as a comparison, what the quality of care looks like in other payment systems during COVID-19 period. In the future, qualitative studies are needed to thoroughly investigate the explanation for our findings. In addition, an assessment of health care professionals’ perceptions is also needed to get a more complete picture of the quality of services. Notwithstanding aforementioned limitations, as the data set used in this study comes from representative primary health care facilities in Poland, the presented results provide a fairly quick assessment of the impact of crisis conditions such as the COVID-19 pandemic on the performance of primary health care providers. Therefore, they can be a guide, globally, for decision-makers during crisis and pandemic situations, as well as for Polish government representatives at the further stages of the reform of primary health care.

The impact of pandemic on different regions is very much alike. The medical personnel have similar odds to get infected or get on quarantine all over the world. The same applies to patients, who prefer to stay at homes, rather than go to clinics where they can get infected. We expect that the transition to remote work and the reduction of inpatient health care in primary care entities will be applied worldwide in the long term. Hence, we believe that the results of this study and a critical assessment of the capitation system, which turns out to be ineffective in crisis conditions such as COVID-19, should be of interest to those responsible for financial management in health care not only in Poland, but also in other countries.

## Figures and Tables

**Table 1 ijerph-18-01407-t001:** The services provided by GPs and nurses in the primary health care system in Poland.

Services of GPs	Services of Nurses
disease prevention, including research and advice on prevention of developmental age and preventive vaccinations, cardiovascular disease prevention, providing advice in the treatment of diseases, including laboratory, imaging and non-imaging diagnostics, performing treatments in the treatment room and at the patient’s home, other services, including: referral to specialist clinics and hospital treatment, referral to rehabilitation.	preventive services, including patronage visits to children from three to nine months of age, and screening tests for children aged 0–6 years old, tuberculosis prophylaxis, performing injections and treatments, diagnostic services, including: collecting materials for diagnostic tests on the basis of an order from a GP, in a situation where the collection for medical reasons should be performed at the patient’s home.

Source: NHF [9].

**Table 2 ijerph-18-01407-t002:** The description of selected to the case studies primary health care services providers.

The Clinics	Location	Other Health Care Services Provided	Primary Health Care Sales as a % of Total Sales of the Clinic	NHF-Financed Sales as a % of Total Sales of the Clinic
Clinic 1	Radom	cardiology, dermatology, rheumatology, gynecology, otorhinolaryngology, ophthalmology	42%	82%
Clinic 2	Warsaw	dentistry, cardiology, dermatology,	43%	79%
Clinic 3	Warsaw	otorhinolaryngology, gastroenterology, ophthalmology, occupational medicine	33%	78%
Clinic 4	Warsaw	ophthalmology, otorhinolaryngology, dentistry	47%	83%

Source: Authors’ own research.

**Table 3 ijerph-18-01407-t003:** Financial indicators of Clinic 1 in March–November 2019 and March–November 2020. All amounts in EUR thousand.

Type of the Service	Sales Revenue, March–November 2019	Sales Revenue Structure, % *	Sales Revenue, March–November 2020	Sales Revenue Structure, % *	Sales Revenue Change, 2019–2020	Sales Revenue Change (%), 2019–2020
GPs—adults	212	55.9%	240	35.9%	28	13.2%
GPs—children	30	7.9%	34	5.1%	4	13.3%
Nurses	49	12.9%	51	7.6%	2	4.1%
Midwifes	13	3.4%	14	2.1%	1	7.7%
School nursery	64	16.9%	88	13.2%	24	37.5%
Diagnostics	10	2.6%	242	36.2%	232	2320.0%
**Sales**	**379**	**100.0%**	**669**	**100.0%**	**290**	**76.5%**
GP—adults	119	31.4%	98	14.6%	−21	−17.6%
GP—children	8	2.1%	17	2.5%	9	112.5%
Nursery	68	17.9%	66	9.9%	−2	−2.9%
Midwifes	10	2.6%	7	1.0%	−3	−30.0%
School nursery	48	12.7%	60	9.0%	12	25.0%
**Salaries**	**253**	**66.8%**	**248**	**37.1%**	**−5**	**−2.0%**
Diagnostics	9	2.4%	118	17.6%	109	1211.1%
Materials	2	0.5%	10	1.5%	8	400.0%
**Medical margin**	**115**	**30.3%**	**292**	**43.6%**	**177**	**153.9%**

*: As a percentage of total sales. Source: Authors’ own research. Lines in bold indicate the sums for sales revenues (lines 1–6) costs of salaries (lines 8–12) and medical margins being the difference between sales and expenditures (salaries, diagnostics, materials).

**Table 4 ijerph-18-01407-t004:** Number of services provided by GPs and nurses from Clinic 1 in the periods March–November 2019 and March–November 2020.

	Total		Monthly Mean		Change		Total
Type of the Service	N° of Services March–November 2019	N° of Services March–November 2020	N° of Services March November 2019	N° of Services March–November 2020	N° of Services, Change, 2019–2020	N° of Services, Change (%), 2019–2020	N° of Services November 2020
GPs services							
Site services	17,371	14,009	1930	1557	−373	−19.3%	1338
Home visits	102	4	11	0	−11	−100%	0
Telemedicine visits	0	1770	0	197	197	N/A	751
Total	17,473	15,783	1941	1754	−187	−9.6%	2089
Nurses services							
ECG	2367	1413	263	157	−106	−40.3%	202
Holter	225	99	25	11	−14	−56.0%	14
Injections	1314	684	146	76	−70	−47.9%	92
Spirometry	90	36	10	4	−6	−60.0%	0
Vaccinations	99	45	11	5	-6	−54.5%	2
Influenza vaccinations	27	72	3	8	5	166.7%	10
Total	4122	2349	458	261	−197	−43.0%	320

Source: Authors’ own research.

**Table 5 ijerph-18-01407-t005:** Financial indicators of primary health care service providers (Clinic 2) in March–November 2019 and March–November 2020. All amounts in EUR thousand.

Type of the Service	Sales Revenue, March–November 2019	Sales Revenue Structure, % *	Sales Revenue, March–November 2020	Sales Revenue Structure, % *	Sales Revenue Change, 2019–2020	Sales Revenue Change (%), 2019–2020
GP—adults	151	69.9%	168	71.2%	17	11.3%
GP—children	22	10.2%	21	8.9%	−1	−4.5%
Nursery	35	16.2%	35	14.8%	0	0.0%
Midwifes	8	3.7%	8	3.4%	0	0.0%
School nursery	0	0.0%	0	0.0%	0	N/A
Diagnostics	0	0.0%	3	1.3%	3	N/A
**Sales**	**216**	**100.0%**	**236**	**100.0%**	**20**	**9.3%**
GP—adults	68	31.5%	55	23.3%	−13	−19.1%
GP—children	7	3.2%	8	3.4%	1	14.3%
Nursery	26	12.0%	21	8.9%	−5	−19.2%
Midwifes	6	2.8%	1	0.4%	−5	−83.3%
School nursery	0	0.0%	0	0.0%	0	N/A
**Salaries**	**106**	**49.1%**	**84**	**35.6%**	**−22**	**−20.8%**
Diagnostics	9	4.2%	5	2.1%	−4	−44.4%
Materials	2	0.9%	3	1.3%	1	50.0%
**Medical margin**	**99**	**45.8%**	**144**	**61.0%**	**45**	**45.5%**

*: as a percentage of total sales. Source: Authors’ own research. Lines in bold indicate the sums for sales revenues (lines 1–6) costs of salaries (lines 8–12) and medical margins being the difference between sales and expenditures (salaries, diagnostics, materials).

**Table 6 ijerph-18-01407-t006:** Number of GPs services and laboratory diagnostics provided to the patients of Clinic 2 in March–November 2019 and March–November 2020.

	Total		Monthly Mean		Change		Total
Type of the Service	N° of Services, March–November 2019	N° of Services, March–November 2020	N° of Services, March–November 2019	N° of Services, March–November 2020	N° of Services Change, 2019–2020	N° of Services Change (%), 2019–2020	N° of Services, November 2020
GPs services							
Site services	9383	9246	1043	1027	−16	−1.5%	834
Home visits	175	59	19	7	−12	−63.2%	6
Telemedicine visits	0	500	0	56	56	N/A	425
Total	9558	9805	1062	1090	28	2.6%	1265
Laboratory diagnostics							
Full biochemistry	6530	4035	6530	4035	−278	−38.3%	959
Haematology	1475	826	164	92	−72	−43.9%	233
Hormones	1047	661	116	73	−43	−37.1%	154
Urine and feces tests	1040	582	116	65	−51	−44.0%	154
Coagulation factors	334	193	37	21	−16	−43.2%	46
Cancer diagnostic	109	80	12	9	−3	−25.0%	21
Autoimmunology	3	14	0	2	2	N/A	0
Total	10,538	6391	1171	710	−461	−39.4%	1567

Source: Authors’ own research.

**Table 7 ijerph-18-01407-t007:** Financial indicators of primary health care service provider (Clinic 3) in a March–November 2019 and March–November 2020. All amounts in EUR thousand.

Type of the Service	Sales Revenue, March–November 2019	Sales Revenue, Structure, % *	Sales Revenue, March–November 2020	Sales Revenue Structure, % *	Sales Revenue Change, 2019–2020	Sales Revenue Change (%), 2019–2020
GP—adults	116	68.2%	132	64.1%	16	13.8%
GP—children	15	8.8%	14	6.8%	−1	−6.7%
Nursery	28	16.5%	30	14.6%	2	7.1%
Midwifes	11	6.5%	12	5.8%	1	9.1%
Diagnostics	0	0.0%	18	8.7%	18	N/A
**Sales**	**170**	**100.0%**	**206**	**100.0%**	**36**	**21.2%**
GP—adults	57	33.5%	54	26.2%	−3	−5.3%
GP—children	2	1.2%	10	4.9%	8	400.0%
Nursery	28	16.5%	41	19.9%	13	46.4%
Midwifes	13	7.6%	2	1.0%	−11	−84.6%
**Salaries**	**99**	**58.2%**	**107**	**51.9%**	**8**	**8.1%**
Diagnostics	7	4.1%	6	2.9%	−1	−14.3%
Materials	3	1.8%	7	3.4%	4	133.3%
**Medical margin**	**61**	**35.9%**	**86**	**41.7%**	**25**	**41.0%**

* as a percentage of total sales. Source: Authors own research. Lines in bold indicate the sums for sales revenues (lines 1–6) costs of salaries (lines 8–12) and medical margins being the difference between sales and expenditures (salaries, diagnostics, materials).

**Table 8 ijerph-18-01407-t008:** Number of services provided by GPs, nurses and the number of laboratory diagnostics provided to the patients of Clinic 3 in March–November 2019 and March–November 2020.

	Total		Monthly Mean		Change		Total
Type of the Service	N° of Services, March–November 2019	N° of Services, March–November 2020	N° of Services, March–November 2019	N° of Services, March–November 2020	N° of Services Change, 2019–2020	N° of Services Change (%), 2019–2020	N° of Services, November 2020
GPs services							
Site services	7446	7794	827	866	39	4.7%	650
Home visits	40	7	4	1	−3	−75.0%	0
Telemedicine	0	502	0	56	56	N/A	457
Total	7486	8303	831	923	92	11.1%	1107
Nurses services							
ECG	941	711	105	79	−26	−24.8%	68
Holter	113	159	13	18	5	38.5%	32
Injections	382	255	42	28	−14	−33.3%	61
Spirometry	0	0	0	0	0		0
Vaccinations	105	160	12	18	6	50.0%	20
Influenza vaccinations	36	93	4	10	6	150.0%	8
Total	1577	1378	176	153	−23	−13.1%	189
Laboratory diagnostics							
Biochemistry	5924	4974	658	553	−105	−16.0%	100
Hematology	1528	1316	170	146	−24	−14.1%	26
Hormones	756	777	84	86	2	2.4%	20
Urine and feces tests	667	516	74	57	−17	−23.0%	21
Conclusion system	249	191	28	21	−7	−25.0%	11
Cancer diagnostic	80	85	9	9	0	0%	2
Autoimmunology	60	37	7	4	−3	−42.9%	0
Total	9264	7896	1030	876	−154	−15.0%	154

Source: Authors’ own research.

**Table 9 ijerph-18-01407-t009:** Financial indicators of primary health care service provider (Clinic 4) in a March–November 2019 and March–November 2020. All amounts in EUR thousand.

Type of the Service	Sales Revenue, March–November 2019	Sales Revenue Structure, % *	Sales Revenue, March–November 2020	Sales Revenue Structure, % *	Sales Revenue Change, 2019–2020	Sales Revenue Change (%), 2019–2020
GP—adults	151	71.9%	148	72.5%	−3	−2.0%
GP—children	12	5.7%	15	7.4%	3	25.0%
Nursery	33	15.7%	30	14.7%	−3	−9.1%
Midwifes	14	6.7%	9	4.4%	−5	−35.7%
Diagnostics	0	0.0%	2	1.0%	2	N/A
**Sales**	**210**	**100.0%**	**204**	**100.0%**	**−6**	**−2.9%**
GP—adults	69	32.9%	75	36.8%	6	8.7%
GP—children	9	4.3%	6	2.9%	−3	−33.3%
Nursery	14	6.7%	23	11.3%	9	64.3%
Midwifes	22	10.5%	9	4.4%	−13	−59.1%
**Salaries**	**113**	**53.8%**	**113**	**55.4%**	**0**	**0.0%**
Diagnostics	11	5.2%	3	1.5%	−8	−72.7%
Materials	2	1.0%	4	2.0%	2	100.0%
**Medical margin**	**85**	**40.5%**	**83**	**40.7%**	**−2**	**−2.4%**

*: As a percentage of total sales. Source: Authors’ own research. Lines in bold indicate the sums for sales revenues (lines 1–6) costs of salaries (lines 8–12) and medical margins being the difference between sales and expenditures (salaries, diagnostics, materials).

**Table 10 ijerph-18-01407-t010:** Number of services provided by GPs and nurses to the patients of Clinic 4 in March–November 2019 and March–November 2020.

	Total		Monthly Mean		Change		Total
Type of the Service	N° of Services, March–November 2019	N° of Services, March–November 2020	N° of Services, March–November 2019	N° of Services, March–November 2020	N° of Services Change, 2019–2020	N° of Services Change (%), 2019–2020	N° of Services, November 2020
GPs services							
Site services	10,866	8886	1207	987	–220	–18.2%	717
Home visits	43	0	5	0	–5	–100%	0
Telemedicine	0	20	0	2	2	N/A	8
Total	10,909	8906	1212	989	–223	–18.4%	725
Nurses services							
ECG	558	234	62	26	–36	–58.1%	12
Holter	0	0	0	0	0	N/A	0
Injections	396	198	44	22	–22	–50.0%	27
Spirometry	144	0	16	0	–16	–100%	0
Vaccinations	36	189	4	21	17	42.0%	14
Influenza vaccinations	18	153	2	17	15	750.0%	14
Total	1152	774	128	86	–42	–32.8%	67

Source: Authors’ own research.

## Data Availability

Data is contained within the article or supplementary material.

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
