# Peer review of "The Impact of COVID-19 on the Performance of Primary Health Care Service Providers in a Capitation Payment System: A Case Study from Poland"

_ijerph, 2021, doi:10.3390/ijerph18041407_

Round 1
Reviewer 1 Report
The paper is very original according to the studied topic. Also, the findings showed: low quality of services provided by primary healthcare in a time of COVID-19 pandemic, inefficient of capitation payment, among others.
Some minor changes must be done:
- in the abstract, delete the titles of the main paragraphs: background, methods, results, conclusions;
- despite being the first study to explore this topic, some references noted in the discussion paragraph could be introduced in the bibliographic framework;
- Finally, the number of tables is excessive. Some of these could be joint. Also, the source of these must be shown below.
Author Response
"Please see the attachment.

Reviewer 2 Report
Commentaries:
- The work entitled "The impact of COVID-19 on the performance of primary health care service providers in a capitation payment system: case study from Poland" has a successful approach, with virtues and some gaps.
- Why is it necessary to carry out this study in Poland? Does the health sector require it? Who can benefit from the results?
- Is it extrapolated to other regions with the same training system or is it limited by the study? In line 10 they indicate "as in many other countries" ... which are these? It is an unsolved detail as soon as you start reading.
- What is the bibliography used for Tables 1 and 2?
- Lines 159-160. "We have not limited our selection to the clinics that operate only inprimary health care industry" ... Why? It is necessary to explain and justify.
- Lines 177-178: "We combined financial data at the level of each practice regarding National Health Fund payments, salary costs and medical margin with administrative data." In what other studies has this practice been used for the reader to consider that it is an accurate and reliable practice and not random?
- Use punctuation in English properly.
- Lines 362-365: "This study shows that in a crisis situation such as the COVID-19 pandemic, capitation funding is associated with a deterioration in the efficiency of the primary healthcare practice as-
sessed on the basis of admission rates and the number of medical services. "Is this a novelty in relation to other studies or is it in the same line? The discussion needs to develop.
Reviewer 3 Report
Comments and Suggestions for Authors
The manuscript (MS) is about an adaptation of healthcare services provided by a case-study composed by four clinics.
Generally, the MS is well written and flows adequately. Its structure is quite simple, but effective though. The Discussion is solid, but the Conclusion seems too long and eventually repeats unnecessarily some points already treated within the Discussion section. Some parts of the Conclusion should be better allocated if in the Discussion section.
The MS does not seem in the need of many amendments. Nevertheless, please consider the following suggestions and comments below that might require some improvement:
Abstract
Lines 13 and 19: The punctuation is not correct. Instead of “… patients; The objective…” and “… measures; Results: We…” should be “… patients. The objective…” and “… measures. Results: We…”, respectively.
- Introduction
L 38: In the sentence “…pandemic. According the Pareto law…” it should be “…pandemic. According to the Pareto law…” instead.
L59-60: In the sentence “Both public or private…” should be “Either public or private...” or “Both public and private…”, but not as it is.
L72 and others therein: The Polish currency should also be given in Euro or US Dollar equivalents and be given a time reference. For example: PLN 80 (apr. € 18 by Decembre 2020 values).
L125: In the sentence “… resulted also in a significant changes in the functioning…” there is a typo that should be amended to “… resulted also in a significant change in the functioning…”.
L126: The sentence “Health services will never be the same again.” should be removed. Despite most of us are aware of that as a given fact, the reason for its removal is because the sentence does not seem based on scientific evidence, but rather on anecdotal cases.
- Material and Methods
L151-153: This sentence seems to be ambiguous and lack of sense. Please rephrase.
Table 2: The third column does not read well. As it is, it confuses the reader. Either provide more space between main rows, or use a dotted line in between, or use alternated greyish stripes as the one used in table 3.
L167-169: In the sentence “The study used administrative and financial data collected in the period from March to November 2019 (pre-COVID-19 period) and from March to November 2020 (post COVID-19 period).” The first assumption seems correct. However, the second assumption is simply not right. COVID-19 is not eradicated yet. So, the second assumption should be changed to something like "first-wave COVID-19 period".
L181-182; 195; and others therein: As referred previously: This assumption should be changed.
E.g., L199 and 201: The word “healthcare” sometimes appears as “health care”. It should be used only in one way.
Table 3: The abbreviation “MD” is not explained neither here nor earlier as “Medical Doctor”. It should.
- Discussion
L446: Please correct the typo to the right author's name: L’Esperance.
Round 2
Reviewer 2 Report
Comments:
- Line 32: delete the point in "care."
- Line 34: indicate the meaning of UNICEF
- It remains undefined what the objective of the work is (not the purpose that they describe it)
- Lines 180-181. Also specify in the Introduction where the data is obtained
- Table 2: the first column does not contain acronyms. They must change the title of the column, for example, by key.
- What is the reason for choosing the months March-November? Why didn't they cover the entire period, that is, March-February?
- Could one of the limitations be the level of reliability of the data?
- Could these results be extrapolated in any way to other bordering regions or are they very particular results of polka health management?
Author Response
Thank you very much for revision of our paper. We have addressed each of your indications. Please find below our comments.
- Line 32: delete the point in "care."
We have deleted this point
- Line 34: indicate the meaning of UNICEF
We have indicated the meaning of UNICEF
- It remains undefined what the objective of the work is (not the purpose that they describe it)
We have changed L 164-166 .
- Lines 180-181. Also specify in the Introduction where the data is obtained
we specified in the Introduction (170-176) where and how the data was obtained
- Table 2: the first column does not contain acronyms. They must change the title of the column, for example, by key.
We have changed the title of the first column
- What is the reason for choosing the months March-November? Why didn't they cover the entire period, that is, March-February?
The pandemic has commenced in Poland in March 2020, therefore the study begins in that month. Since the article has been written in December 2020 the period of the study ends in November 2020. In order to study the impact of Covid-19 on performance of primary healthcare providers under capitation system we compared their performance in aforementioned period with their performance in the same period of 2019.
- Could one of the limitations be the level of reliability of the data?
Given the study provides the very first insight on the impact of the pandemic on primary healthcare systems we consider the following papers will provide more results and conclusions. However, since this study employs mixed methods, i.e. data analysis and interviews and that we obtained aligned results from both these methods, we consider the results are reliable.
- Could these results be extrapolated in any way to other bordering regions or are they very particular results of polka health management?
We have added explanation in L 625 - 633